# Oral–Gut Microbiota and Arthritis: Is There an Evidence-Based Axis?

**Lorenzo Drago [1,\*], Gian Vincenzo Zuccotti [2], Carlo Luca Romanò [3,4], Karan Goswami [5], Jorge Hugo Villafañe [6], Roberto Mattina [7] and Javad Parvizi [5]**

1 Laboratory of Clinical Microbiology, Department of Biomedical Sciences for Health & Microbiome, Culturomics and Biofilm related infections (MCB) Unit, "Invernizzi" Pediatric Clinical Research Center, University of Milan, 20133 Milan, Italy

2 Department of Pediatrics, V. Buzzi Childrens' Hospital & "Invernizzi" Pediatric Clinical Research Center University of Milan, 20141 Milan, Italy; gianvincenzo.zuccotti@unimi.it

3 Carlo Luca Romanò, Studio Medico Cecca-Romanò, Corso Venezia, 2, 20121 Milano, Italy; carlo.romano@unimi.it

4 Romano Institute, Rruga Ibrahim Rugova, 1, 00100 Tirane, Albania

5 Rothman Institute, Thomas Jefferson University, Philadelphia, PA 89814, USA; karan.goswami@mail.com (K.G.); javadparvizi@gmail.com (J.P.)

6 IRCCS Fondazione Don Carlo Gnocchi, 20141 Milan, Italy; mail@villafane.it

7 Department of Biomedical, Surgical and Dental Science, University of Milan, 20133 Milan, Italy; roberto.mattina@unimi.it

\* Correspondence: lorenzo.drago@unimi.it

**Abstract:** The gut microbiome appears to be a significant contributor to musculoskeletal health and disease. Recently, it has been found that oral microbiota are involved in arthritis pathogenesis. Microbiome composition and its functional implications have been associated with the prevention of bone loss and/or reducing fracture risk. The link between gut–oral microbiota and joint inflammation in animal models of arthritis has been established, and it is now receiving increasing attention in human studies. Recent papers have demonstrated substantial alterations in the gut and oral microbiota in patients with rheumatoid arthritis (RA) and osteoarthritis (OA). These alterations resemble those established in systemic inflammatory conditions (inflammatory bowel disease, spondyloarthritides, and psoriasis), which include decreased microbial diversity and a disturbance of immunoregulatory properties. An association between abundance of oral *Porphyromonas gingivalis* and intestinal *Prevotella copri* in RA patients compared to healthy controls has been clearly demonstrated. These new findings open important future horizons both for understanding disease pathophysiology and for developing novel biomarkers and treatment strategies. The changes and decreased diversity of oral and gut microbiota seem to play an important role in the etiopathogenesis of RA and OA. However, specific microbial clusters and biomarkers belonging to oral and gut microbiota need to be further investigated to highlight the mechanisms related to alterations in bones and joints inflammatory pathway.

**Keywords:** microbiota axis; gut microbiota; oral microbiota arthritis; joint inflammation

---

## 1. Introduction

*Homo sapiens* is more prokaryotic than eukaryotic, as the bacteria "layed" in the internal mucosae (intestinal tract, reproductive organs, and respiratory tract) and externally in the body (skin and hair) outnumber host cells 10 to 1 [1]. This paradigm shift has been prompted by the advent of high-throughput metagenomic approaches and has definitively changed the way we study human

microbial ecosystems and their interactions with the host. Microbes present in these biological systems are deeply integrated in our daily life, and emerging research has sought to decipher this complex inter-kingdom communication network present in our body and immune system. The gastrointestinal (GI) tract has the highest density and variety of microorganisms (more than 100 trillion microbes and approximately 1500 species). Early life host–microbe interactions, especially in the gut, drive the development of immunity and the establishment of a stable complex microbial community, commonly referred to as the commensal microbiota [2,3]. Extensive research has focused on gut microbiota and host immune response effects in the context of protection against pathogenic gut microbes and the pathophysiology of chronic inflammatory/autoimmune disease states [4,5]. For example, it has been reported that in patients with Crohn's disease, there is a relationship between dysbiosis and response to treatment. Hence, microbiota could be a target of the treatment of chronic intestinal diseases [6].

Emerging scientific reports have also highlighted the immunomodulatory effects of gut microbiota on other pathologic conditions, which often involve distant anatomical sites, such as the liver, the brain, the heart and the skeleton [7–9].

Furthermore, several mechanisms and factors have been implicated to explain the role of microbiota in bone and joint health [10]. The gut microbiome is indeed a source of a number of key vitamins, such as cobalamin (B12), biotin (B7), folate, thiamine (B1), pyridoxal phosphate, pantothenic acid (B5), niacin (B3), vitamin K, and tetrahydrofolate, which are particularly important for the health of the musculoskeletal system [11].

Steves et al. highlighted how the gut microbiome can alter the inflammatory state of an individual by influencing both the host metabolic potential and its innate and adaptive immune system [12]. These authors further discussed the role of microbiota diversity on some prevalent age-related disorders, such as osteoporosis, osteoarthritis, gout, rheumatoid arthritis, frailty and sarcopenia.

In the last decade, the alteration of gut microbiota has been reported in rheumatic disease and arthritis, most notably in juvenile idiopathic arthritis (JIA), rheumatoid arthritis (RA), psoriasis, and the related spondyloarthritides (SpA), including ankylosing spondylitis (AS) and reactive arthritis (ReA) [13]. In a similar fashion to inflammatory bowel disease (IBD), it has been suggested that gut bacteria play important role in the etiopathogenesis of these aforementioned conditions.

RA is an autoimmune disorder which occurs when the immune system affects the fluid that nourishes the cartilage and lubricates the joints (synovium) and their soft tissues. Generally, the root causes of arthritis include an increase in inflammatory processes and a decrease of the normal amount of cartilage present at the joint. A correct diet and gut balance can improve these diseases [14]. Indeed, inflammation-reducing foods containing antioxidants, such as fresh fruits, vegetables, or a gluten-poor diet may improve symptoms and disease progression by restoring intestinal microbiota. Findings have provided a model of how genetic and environmental factors, in association, cause autoimmune diseases such as RA. Sakaguchi S. et al. reported that the causal genetic anomaly of *ZAP-70*, a polymorphism of the *MHC* gene, significantly contributes to determining genetic susceptibility to autoimmune arthritis in SKG mice. Furthermore, they demonstrated that the disease initiation requires the interaction of both genetic and environmental factors, in particular the type of microbial colonization.

One of the most common form of arthritis is osteoarthritis (OA). This disease commonly occurs when the protective cartilage on the ends of bones wears down over time by damaging any joint of the hands, knees, hips and spine. OA is characterized by a chronic, low-grade inflammation which is mediated primarily by the innate immune system, making it distinct from that observed in RA. Several dietary factors have been reported to be involved in the pathogenesis of OA. Vitamins, magnesium, and especially amino acids, i.e., little amounts of single amino acids supplementation such as 0.5% (w/w) ʟ-arginine or 1.0% (w/w) ʟ-glutamine, have shown a significant influence on intestinal microbiota, especially the ratio of Firmicutes/Bacteroidetes. Chitosan supplementation can also alter the component of intestinal microbiota, causing a lowering of the ratio Firmicutes/Bacteroidetes, in particular a decreasing of Bacteroidales and an increasing of the Lactobacillales in the feces [15,16].

The alteration of gut microbiota can thus lead to an increased translocation of microbial associated molecular patterns (MAMPs) across the gut endothelium into the systemic circulation. MAMPs include factors such as lipopolysaccharide (LPS), peptidoglycan, and bacterial DNA. These factors can trigger pro-inflammatory pathways by stimulating immune receptors in the resident immune cells of bone, cartilage and synovium [17,18].

RA has long been associated with periodontal disease [19], and recent evidence on the oral microbiome has emphasized its role in the arthritis. Using a metagenomics approach and molecular investigations, common opinion has been formed that each individual carries over 700 species in the oral cavity, and this microbiome is the second largest microbial niche after the gastrointestinal tract [20]. Oral bacteria may penetrate through the gingival pockets and enter into the bloodstream. The translocation of microbiota-derived molecules into systemic circulation is considered one route for the microbiome to mediate arthritis by stimulating specific cytokines (see below).

There is not so much evidence on microbiota association with some musculoskeletal diseases related to age, as RA and OA. However, it seems that these clinical issues are associated with inflammatory changes, which could be specifically related to microbiota changes or be associated with age. Some studies described below on microbiota and arthritis were age-matched, presuming that the shaping of microbiota may have a role in the developing and maintaining these diseases independently by age.

The present review aims to address the most recent findings regarding the oral and gut microbiomes and their relationship with RA and OA, respectively.

## 2. Oral Microbiota in RA and OA

RA is an autoimmune disease affecting the synovium and cartilage with bony erosion. Recently, the relationship between the oral microbiome and systemic diseases has been explored [21,22]. Sher et al. demonstrated that overall exposure to *Porphyromonas gingivalis* was similar between patients with RA and controls. These authors found an abundance of *Anaeroglobus geminatus* that correlates with the presence of rheumatoid factors, and *Prevotella* and *Leptotrichia* species are the only taxa that have been observed in patients with new-onset RA [22].

Chen et al. showed that RA has a distinct oral microbiome and may be affected by its dynamic variations [23]. In this study comparing the oral microbiome in RA, OA and healthy patients using rRNA gene amplicon sequencing, eight oral bacterial biomarkers (*Prevotella melaninogenica*, *Veillonella dispar*, *Prevotella*, *Neisseria*, *Porphyromonas*, *Veillonella*, *Haemophilus*, *Rothia*, *Streptococcus*, *Actinomyces*, *Granulicatella*, *Leptotrichia*, *Lautropia*, and *Fusobacterium)* were identified to differentiate RA from OA. In addition, the authors found that patients with RA and OA had oral microbiota with higher microbial diversity compared to healthy subjects, indicating that there could be more pathobionts in the oral cavity of patients with RA that are able to negatively influence the outcome of the disease. The most common phyla were Proteobacteria, Firmicutes, Bacteroidetes, Actinobacteria and Fusobacteria. The relative abundance of Proteobacteria in healthy subjects was significantly higher than in patients with RA and OA, and the relative abundance of Firmicutes in patients with OA is significantly higher than those in patients with RA. Table 1a,b reports the different taxa and species of oral and gut microbiota observed in RA and OA, respectively.

Persson et al. previously noted *P. gingivalis* to be directly linked to RA through citrullination and induction of antipeptidyl citrulline antibodies reacting to citrullinated human self-proteins [24]. Interestingly, *P. gingivalis*, which is mainly abundant in the oral microbiome of RA patients, shares 82% homology of $\alpha$-enolase with human $\alpha$-enolase. Consequently, human antibodies against bacterial enolase can promote an increase of antibody production. Lundberg et al. [25] showed that the levels of anti-citrullinated human $\alpha$-enolase antibodies and bacterial $\alpha$-enolase correlates with the severity of RA. *P. gingivalis* can be also found in the gut, but nothing exists between the *P. gingivalis* oral–gut axis correlation and arthritis, and the presence of this bacteria in the gut is not an inflammatory trigger of RA.

**Table 1.** Differences of bacterial abundance (taxa) in oral and gut microbiota of rheumatoid arthritis (RA) (**a**) and osteoarthritis (OA) (**b**) patients compared to the healthy controls.

| (a) | | | | |
|---|---|---|---|---|
| **Type of Arthritis** | **Abundance** | **Oral Microbiota Profile (Taxa)** | **Abundance** | **Gut Microbiota Profile (Taxa)** |
| RA | **Increase** ([23]) | *Neisseria subfava, Haemophilus parainfuenzae, Veillonella dispar, Prevotella tannerae, Actinobacillus parahaemolyticus, Neisseria, Haemophilus, Prevotella, Veillonella, Fusobacterium, Aggregatibacter, Actinobacillus* | **Decrease** ([26,27]) | *Bacteroides, Akkermansia, F.prausnitzii Prevotella, Ruminococcus* |
| RA | **Increase** ([24]) | *Porphyromonas gingivalis* | | |
| RA | **Decrease** ([28]) | *Haemophilus* spp. | **Decrease** ([28]) | *Haemophilus* spp. |
| RA | **Increase** ([28]) | *Lactobacillus salivarius* | **Increase** ([28]) | *Lactobacillus salivarius* |
| RA | **Increase** ([29]) | ***Plaque:*** *Actinomyces meyeri Prevotella nigrescens Treponema socranskii Treponema* spp. *Eubacterium infirmum Prevotella oris Actimomyces massiliensis Catonella* spp. | **Increase** ([30,31]) | *Prevotella copri* |
| RA | **Increase** ([29]) | ***Saliva:*** *Prevotella* spp. *Veillonella* spp. *Centipeda* spp. *Solobacterium morei Prevotella pallens Atopobium parvulum Butyrivibrio* spp. | **Decrease** ([30]) | *Bacteroides* |
| RA | **Increase** ([32]) | *P. melaninogenica P. denticola P. histicola, P. nigrescens, P. oulorum P. maculosa Selenomonas noxia S. sputigena Anaeroglobus geminatus Aggregaticbacter actinomycetemcomitans Parvimonas micra Other Gram-negatives* | | |

**Table 1.** *Cont.*

| (a) | | | | |
|---|---|---|---|---|
| **RA** | **Decrease** ([32]) | *Streptococcus Rothia aeria Kingella oralis Haemophilus Actinomyces* | | |
| **(b)** | | | | |
| **OA** | **Increase** ([28]) | *Rothia dentocariosa, Ruminococcus gnavus, Streptococcus, Actinomyces, Lautropia, Rothia, Granulicatella, Ruminococcus, Oribacterium, Abiotrophia* | **Increase** ([33]) | *Lactobacillus* spp. *Methanobrevibacter* |
| **OA** | | | **Increase** ([33]) | *C. coccoides, C. leptum, Clostridium clusters XI-I, Roseburia* spp., *Lactobacillus* spp. |
| **OA** | | | **Decrease** ([33]) | *Bacteroides Prevotella* spp. |

Eriksson et al. [29], by investigating the periodontal health of patients with RA in relation to oral microbiota and inflammatory levels, found that the majority of the patients had moderate or severe periodontitis and a higher production of anti-citrullinated protein antibodies. The microbiota observed in the plaque were different compared to the saliva samples. The relative bacterial abundances in both sites are shown in Table 1a,b.

A very recent study [32] characterized the subgingival microbiome of RA patients and its association with periodontal severity. The authors demonstrated that changes in the oral microbiota, especially of those species associated with periodontal disease, were linked to worse RA. The abundance of *Prevotella* spp. and the reduction of health-associated species (*Streptococcus*, *Rothia*) may cause an increased production of inflammatory mediators including IL17, IL-2, TNF, and IFN-γ.

Microbial oral translocation into the systemic bloodstream is considered a negative pathway to induce a systemic pro-inflammatory trigger. A recent study reported that the systemic diffusion of bacterial lipopolysaccharide (LPS), a cell wall compound of gram-negatives bacteria, positively correlated with joint inflammatory response and the severity of joint degradation [17]. LPS can also be concentrated into the synovial fluids and upregulate specific pro-inflammatory cytokines. These immunological factors can have an important role in the pathogenesis of arthritis, especially in RA [34]. It is thus probable that many other bacterial clusters and biomarkers can be involved in the increasing of those local or systemic inflammatory conditions which lead to joint/cartilage damage and corrosion.

As mentioned, a clear correlation between bacteria and OA can also be seen by studying the profile of the oral cavity. Oral microbiota seem to have a particular value in OA as well as in the differentiation of RA. Despite these challenging results, more in-depth studies are needed to explore the differences in the oral microbiome profiles of patients with OA. Next-generation sequencing may be a useful tool to further investigate how oral bacteria can affect this type of arthritis.

### 3. Gut Microbiota in RA and OA

The hypothesis that not only oral but also intestinal microbiota can be associated with the development of RA is supported by many data. Zhang et al. published a case-control metagenome-wide association study (MGWAS) of the fecal, dental and salivary microbiomes of a cohort of treatment-naive and treated RA patients. They found that the RA-associated microbiome deviated significantly from healthy controls in all sites [28]. In this study, *Haemophilus* spp. was depleted in individuals with RA at the fecal and oral levels, whereas *Lactobacillus salivarius* was over-represented in individuals with RA at both microbiota sites.

Older patients often manifest more severe diseases, and this appears connected to age-related gut dysbiosis. Alterations in the microbiota provide plausible candidate mechanisms for driving both inflammation and altering the immune response and host metabolism, which in turn may modulate the development of musculoskeletal problems (see the *Prevotella copri* case below). The microbiome is thus a highly plausible target for the modulation of diseases of aging owing to its close relationship with innate and adaptive immune systems. Components of intestinal microbiota can indeed direct key aspects of host immunity, in particular effector T-cell differentiation, which may impact susceptibility to autoimmune diseases and RA in particular [35].

Different studies investigating the etiology of RA have established the involvement of regulatory T (T-reg) cells, which are defective at suppressing IFN-$\gamma$ and TNF-$\alpha$ production by conventional T cells in the peripheral blood of active RA patients [36,37]. It has been well established that the gut microbiota–immune interaction and homeostasis, via balancing pro- and anti-inflammatory mechanisms, can regulate the differentiation of various T cell types, especially T-reg cells [38]. A clear example is the potential therapeutic effect of SCFAs (short chain fatty acids), which are microbial fermentation products found in the bowel, that have demonstrated a profound influence on T-reg cell differentiation in a variety of experimental models of autoimmunity or inflammatory T-cell-mediated diseases [39,40].

An elegant collagen-induced arthritis mouse model published by Hui et al. demonstrated that butyrate (a functional SCFA) administration inhibited arthritis by suppressing the expression of inflammatory cytokines [41]. The authors suggested that modulation was likely mediated by the differentiation of CD4 T cells towards T-reg cells, which produce anti-inflammatory cytokine IL-10, and thus influenced the function of Th17 cells.

As mentioned, an altered microbiota profile has also been associated with juvenile idiopathic arthritis (JIA). Current evidence indeed suggests that the perturbation of gut microbiota may contribute to the development of JIA [42,43].

It remains difficult, however, to establish a definitive microbial marker or specific enterotypes that are associated with RA. It has been hypothesized that the alteration of single bacterial genus could have direct impact on driving inflammation, as suggested for *Bacteroides*, *Akkermansia* or the anti-inflammatory *Faecalobacterium prausnitzii*, which has been noted to be depleted in RA patients, while *Prevotella* and *Ruminococcus* were more prevalent [26,27]. Increasing *Prevotella copri* and decreasing *Bacteroides* concentrations in the stool have both been associated with new onset, untreated RA in humans [30].

The above studies, although not always homogenous, have directly or indirectly demonstrated that genetic risk may be modulated by alterations in the microbiome and that the presence of particular microbial markers can be predictive of disease. As mentioned, intestinal microbiota are also known to change with age. Many of the clinical issues, including OA, are related to the inflammatory change—either specific to disease or associated with age. OA is indeed considered a degeneration of joint cartilage and the underlying bone which commonly occurs from middle age onward. The precise etiology of OA remains unknown thus far, even if various risk factors have been associated with presence of the disease, including age, sex, obesity, and diet, and local joint injury [44].

Most of these factors are associated with drastic changes in the intestinal microbiota. Age, in fact, shifts the intestinal microbiota with significant differences between younger adults and older people,

showing a lower diversity of gut microbiota, a greater proportion of *Bacteroides* spp., and a distinct abundance pattern of *Clostridium* groups [45]. In addition, obesity is associated with phylum-level changes in the microbiota (i.e., ratio of Firmicutes/Bacteroidetes), reduced bacterial diversity, and an altered representation of bacterial genes and metabolic pathways [46]. Finally, diet can shape gut microbiota and consequently change the composition and metabolism of intestinal microbiota as well as impact host immune responses [47].

For these reasons, OA is now considered an induced inflammatory condition in which the role of the microbiome has emerging as one of the most important factors. Several publications have reported a clear demonstration of the link between osteoarthritis and gut microbiota. For instance, animals with a low-grade chronic systemic inflammation due to a high-fat diet have developed osteoarthritis, and others with an increased body weight due to diet have shown an increased progression of osteoarthritis [48,49]. Metcalfe et al. proposed that metabolic endotoxemia (raised LPS blood and synovial concentrations) caused by impaired gastric mucosa and low-grade chronic inflammation, may contribute to the onset and progression of OA in obese patients [50].

Collins et al. also demonstrated that changes in the Mankin score (a histopathological classification of the severity of osteoarthritic cartilage lesions) seen in a rat model of osteoarthritis were correlated with alterations of gut microbiota [33]. The translocation of bacteria or related compounds (i.e., LPS and peptidoglycans) across the gut barrier into the systemic circulation was found to mediate osteoarthritis. Together, *Lactobacillus* species and *Methanobrevibacter* spp. abundance have shown a strong predictive relationship with the Mankin Score ($p < 0.001$).

Huang and colleagues further demonstrated that systemic and synovial concentrations of bacterial LPS were positively correlated with the joint inflammatory response [17]. This study enrolled 25 patients in whom ostephyte score, joint space narrowing, and pain were measured.

Th epolymerase chain reaction (PCR) analyses and next generation sequencing (NGS) of osteoarthritic synovial fluid and synovial tissue have also revealed the presence of bacterial DNA, raising the possibility that live bacteria or bacterial products are present in the joint during disease progression [51,52].

Other studies have delineated the use of specific probiotics to rebalance gut microbiota and reduce the grade of inflammation. Studies in OA animal models have demonstrated that the oral administration of *Lactobacillus casei* with type II collagen and glucosamine as prebiotic reduces pain, cartilage destruction, and lymphocyte infiltration and leads to a reduced expression of numerous pro-inflammatory cytokines and matrix metalloproteinases, as well as an upregulation of anti-inflammatory cytokines IL-10 and IL-4 [53]. The results observed after the oral intake of a combination of *Lactobacillus casei* and *Lactobacillus acidophilus* in a rat model of collagen-induced arthritis seemed even more protective versus those after indomethacin administration, with regard to oxidative stress parameters in synovial effusate and arthritis scores [54]. A very recent study conducted in a rat model with OA demonstrated that a probiotic diet plus chondroitin sulfate administration reduced the expression of the markers of inflammation and collagen degradation [55].

The exact role of gut microbiota's involvement in the pathophysiology of OA remains under investigation; all these aforementioned observations raise the possibility that the microbiome or part of it may mediate the effects and outcome of this highly prevalent and widespread disease.

## 4. Discussion

The first description of the possible involvement of microbiota in the pathology of arthritis was published in 1970s when rats raised in germ-free conditions developed severe joint inflammation with 100% penetrance in an adjuvant-induced arthritis model, while conventionally raised controls showed only mild disease at a very low incidence [56].

A fine equilibrium between 'peace-keeping' and potentially pro-inflammatory intestinal and oral bacteria is necessary to keep gut immunity in check and prevent a state of dysbiosis, which might lead to local and distant deleterious consequences in the host. A crucial driver of changes in the gut

and oral environments is the inflammatory response of the host. Intestinal and oral inflammations in people are associated with an imbalance in the microbiota, the dysbiosis, which is characterized by a reduced diversity of microbes, a reduced abundance of obligate anaerobic bacteria, and an expansion of facultative anaerobic bacteria in the phylum Proteobacteria, mostly members of the family Enterobacteriaceae.

In regards to RA and gut microbiota, single microorganisms such as *P. copri* might correlate with the development of RA. Pianta A. et al. reported massive concentrations of antibodies against *P. copri* in the sera of RA patients [31]. Impressive advances in sequencing technologies, compelling animal data, and mounting human evidence have suggested that gut microbiota indeed play a part in the pathogenesis of diseases such as autoimmune arthritis. The few studies addressing potential links between the gut microbiota and human inflammatory joint disease have identified dysbiotic patterns that may contribute to initiate or to perpetuate the disease. Obviously, age can greatly contribute to the increase of systemic inflammation (inflammaging), and the microbiota shaped by aging can negatively modulate the outcome of joint diseases. However, the gut microbiota of RA patients seem to be more dysbiotic than those of healthy patients, thus confirming their role as independent of age. An indirect demonstration of the role of microbiota is that gut microbiome (the same for the oral) is perturbed in rheumatoid arthritis and partly normalized after RA-specific treatment [28].

Dysbiosis occurring, for instance, in jejunoileal-bypass, used as alternative to bariatric banding, seems to be associated with arthritis. In these patients, studies have reported a bacterial overgrowth and a deposition of resultant immune complexes in the synovium [57]. However, a very comprehensive human model fitting with the gut–joint axis and dysbiosis can be represented by Whipple's disease, in which the presence of a single bacterium, *Tropheryma whipplei,* overgrowth in the small intestine is sufficient for the development of joint inflammation in predisposed individuals. Similar results have been appointed by some authors regarding the high quantity of *Streptococci* in milk as a theoretical cause of RA [58,59].

A strong evidence of the gut–bone axis and its role in arthritis outcomes has been reported in germ free mice studies. It has been evidenced that these animals do not show arthritis; however, the mono-colonization of particular intestinal bacterium is sufficient to induce arthritis. Thus, gut microbiota have been further confirmed to be a cause of relevant immunological triggers occurring in arthritis pathogenesis [8,60].

Periodontal disease also correlates with new-onset RA patients, and many bacterial clusters related to this disease have been faced in different studies [22–24]. Gene sequencing studies have investigated the subgingival microbiome of patients with RA and compared the results of subjects with osteoarthritis and healthy controls with or without periodontitis. In both cases, literature revealed that specific bacteria biomarker abundance may influence the severity of the osteoarthritic disease. Unfortunately, no unique microbial oral cluster has been identified so far.

Only one study [28] has reported results on the simultaneous effect of oral–gut microbiota in RA patients. By collecting fecal, dental and salivary samples in a cohort of RA and healthy donors, this study demonstrated a rate of dysbiosis in the gut and oral microbiomes of RA patients, suggesting an overlap in the abundance and function of species at different body sites that could be partially resolved after RA treatment.

Despite findings which are suggestive of microbiota–bone axis correlation with inflammatory joint disease, research to date remains inconclusive with regard to the final mechanism. We therefore need to identify the priorities for research in order to substantiate and translate these findings. An important and recent review analyzing nine clinical studies [61] compared changes in diversity and taxa present in the microbiome of RA patients with age, gender and weight-matched controls. Despite microbiome diversity being a generic tool to define whether microbial disturbance in the oral or gut environments has occurred, the study of specific bacterial clusters is of great interest to establish the possible etiopathogenetic role of microbiota in arthritis. In RA, a correlation between a pro-inflammatory genotype-HLA related bacteria and some bacterial clusters has been strongly postulated. However,

well-defined human studies using NGS and metabolomic approaches are needed to better understand if and when intestinal community composition in patients with joint inflammation differs (in addition to improving therapies) by looking at specific bacterial markers for disease presence and progression. Prospective studies evaluating the microbiome–host relationship are indeed necessary to establish not only the potential etiology but also the effects of immunosuppressive or anti-inflammatory therapies on microbiota. Another final aim will be to establish how the microbiota can influence therapies per se in OA or RA patients and, subsequently, how they may impact the host's well-being. Table 1a,b shows the main taxa abundances in oral and gut microbiota in OA and RA. To date, interesting and exhaustive data have shown that a connection between microbiomes and joint diseases exists in RA. Other diseases, OA in particular, have received little attention so far, despite some promising, suggestive findings. The gut microbiome, indeed, could be the culprit behind arthritis and joint pain for obese people. A recent paper by Schott E.M. et al. [62] demonstrated that changes in the gut microbiomes of the mice coincided with signs of body-wide inflammation, including in their knees, where the authors induced osteoarthritis with a meniscal tear. Compared to lean mice, osteoarthritis progressed much more quickly in the obese mice, with nearly all of their cartilage disappearing within 12 weeks of the tear.

Though studies have specifically investigated the influence of gut microbiota in OA, pre-clinical data and some observational investigations in humans have suggested a potential relationship between the gut and risk factors of OA. The role of some confounding factors (genes, sex, age, diet, living conditions) needs to be better explored to fully understand the role of gut bacterial biomarkers in OA.

Thus, a deeper understanding of the biological complexities of our 'two genomes' (host and microbial) will help elucidate the factors that trigger inflammation and finally bridge the gap in our knowledge regarding the role of gene–environment interactions in other autoimmune and inflammatory processes involved in disease pathogenesis. Next generation sequencing, metatranscriptomic analysis, and metabolomic approaches may provide yet-greater insight and help to further understand these mechanisms.

There is a justified association between oral and gut microbiomes in arthritis, although the current evidence that the microbiome causes arthritis is far from conclusive. Strategic future studies aiming to improve the understanding of the combined role of gut–oral axis in arthritis as well as the use of "microbiome influencers," such as the probiotics, are mandatory.

## 5. Highlights of Future Perspectives

Boxes 1–5 report Microbiome definition and its involvement in RA and OA as well as the need for further studies.

**Box 1.** Microbiome definition.

The microbiome is defined as the totality of microorganisms and their genes inhabiting a unique environment; the human microbiome outnumbers human genes by several orders of magnitude.

**Box 2.** Tools for studying microbiome.

Understanding of the role of microorganisms in modulating health and disease by NGS and metabolomic technologies will be the new era.

**Box 3.** Microbiome and RA link.

Despite the fact that precise causation of RA has not yet been established, several clinical investigations have demonstrated the role of some microorganisms in RA pathogenesis, independently of age.

**Box 4.** Microbiome and OA link.

OA is the most common disorder of the musculoskeletal system. The literature has considered the microbiome and the use of some selected probiotics as a possible future therapeutic approach.

**Box 5.** Need for further studies.

More studies are needed to assess the role of the microbiome in human arthritis and related diseases in the order to finally elucidate their mechanisms and therapeutic targets.

**Author Contributions:** L.D. conceived and write the paper; G.V.Z., C.L.R., R.M., J.H.V. revised the paper; K.G. and J.P. revised the English and improved the manuscript.

**Funding:** This research received no external funding

**Conflicts of Interest:** The authors declare no conflict of interest

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
