# Peer review of "Oral–Gut Microbiota and Arthritis: Is There an Evidence-Based Axis?"

_jcm, doi:10.3390/jcm8101753_

Round 1
Reviewer 1 Report
The authors aimed to review relationship with the oral and gut microbiome and arthritis, especially RA and OA. They described the role of oral and gut microbiota for developing and exacerbating RA or OA including molecular mechanisms. They referred many reports and finally commented that the interaction of host and microbial genome could be an important trigger on inflammatory diseases. The author's review is very interesting; however, I have several major and minor concerns.
Major concern:
1. It is better to add some sentences about genetic background such as HLA with microbiota because there are some differences in disease prevalence such as gender, ethnicity. As related to genetic background, I recommend that the authors discuss the paper of SKG mice with ZAP70 mutation, written by Sakaguchi et al.
2. In association with Major concern no.1, I am afraid that the part of “OA and microbiota” did not have a great impact. As the authors also described, the molecular mechanism bridging microbiota and clinical aspects of OA is still unclear. If possible, it is better to discuss those problems in detail, even in speculation.
Minor concerns:
1. In total, each paragraph is too short. For example, in page 2, line 52-54, there was only one sentence.
2. In page 2, line 76 and 79; the authors describe the report about OA in discussing RA.
3. In page 7, line 301-303; it is needed to describe a specific reference number.
4. Throughout the paper, English should be revised.
Author Response
Dear Reviewer, we are grateful for the suggestions and comments, which will have surely improved the quality of the review. Here attached the file as response to the queries and the revised manuscript with the changes addressed accordingly.
My best regards.
Reviewer 2 Report
Line 141, the more description about the RA and microbiota should be don In the part of introduction, no the related description about OA? 1-2 figures should be done to summarize the recent finding about microbiota in RA and OA. The pathogenesis of autoantibodies and RA and microbiota, environmental factors should be presented by a figure. Adequate editing Table 1 should be done Many small paragraphs was observed in this paper, and these paragraphs should be concordance
Author Response

(The authors gave the same response as above.)

Round 2
Reviewer 1 Report
I think that the authors carefully revised their manuscript.
Author Response
Thank you for your support and the acceptance of our revisions.
Best Regards.
Reviewer 2 Report
There was no a good table or figure to increase the readability of this review article. Many small paragraphs were still observed in this paper, and this problem should be corrected.
Author Response
Dear Reviewer, thank you for your considerations. According to your suggestions the Table has been totally renewed and splitted in two parts, a and b, for a better comprehension.
The manuscript includes the following paragraphs:
Introduction (almost 2 pages);
Oral Microbiota in RA and OA (more than 1 page);
Gut Microbiota in RA and OA (more than 2 pages);
Discussion (more than 2 pages);
5 Boxes (Highlights);
1 Table splitted in two (a and b).
References (61 articles)
We hope we have now clarified and satisfied all the raised concerns.
Best Regards.
This manuscript is a resubmission of an earlier submission. The following is a list of the peer review reports and author responses from that submission.
Round 1
Reviewer 1 Report
line 76) clarify the concepts of corretect diet and gut balance
line 102) microbiota change with age and it is related with more severe disease.it is a important information that should be more discussed.
line 125 tryptophane should not be called as a intestinal compound, since it is aminoacid
line 138: use microbiota instead of bacterial flora
Author Response
We would like to thank Referees for their comments. The revised performed will for sure increase the quality of the manuscript.
Best regards.
Reviewer 2 Report
This report showed that gut microbiota in RA patients and OA patients were altered. This is timely review article. However, there are many issues which authors need to clarify. The reviewer could not understand the point that “dysbiosis observed in RA and OA” derived from their old ages or the inflammatory disease condition from this paper. In addition, authors need to add more detailed information about the role of gut microbiota in RA.
Major comments:
Authors need to summarize previous reports which showed dysbiosis was observed in RA and OA in tables, not only in the manuscript. In addition, what are the similar microbiota patterns in RA and OA patients? What is the difference between RA and OA? Authors need to describe these points clearly in the text and tables.
In this field, it has been reported that germ free mice did not show arthritis, however mono-colonization of particular intestinal bacterium is sufficient to induce arthritis. Thus, gut microbiota is thought to be an important trigger of arthritis. Authors need to describe these matters and include the following manuscript. (Wu HJ, et al. Immunity 2010; 32: 815-827.) (Abdollahi-Roodsaz S, et al.The Journal of clinical investigation 2008; 118: 205-216.)
Authors mentioned “Older patients often manifest more severe disease, and this appears connected to age-related oral and gut dysbiosis.” Did authors speculate that altered composition of microbiota in RA was observed because they are in old ages? Most of the case-control studies are age-matched. Therefore, the reviewer could not understand what the author would like to explain from this part. Authors need to change this part.
Minor comments:
In introduction, authors described” Symptoms such as joint pain, stiffness and swelling are typically observed at any age, but commonly seen in adults over the age of 65”. These are the major symptoms in RA, not only in elderly RA. Therefore, authors need to delete this part.
Authors mentioned Prevotella copri might correlates with development of RA. Authors need to include the following paper. Antibody-against P. copri was detectable in serum of RA patients (Pianta A, et al. Arthritis & rheumatology 2017; 69: 964-975.)
Authors showed oral microbiota is changed in RA patients in ‘Gut microbiota and RA’ section. Authors need to show ‘Oral microbiota and RA’ and ‘Gut microbiota and RA’, separately.
JIA is different from RA. Authors should not include these reports in “Gut Microbiota and RA” section.
Author Response
We would like to thank the Referees fo the comments and suggestions. This revised form is surely improved thanks to their support.
Best Regards
Round 2
Reviewer 2 Report
The manuscript was improved. However, the authors could not properly respond to the reviewer’s comments. Moreover, the authors copied some of the phrases of original article.
In line 110-112, authors copied the phrases of the original article “RA and OA had an oral microbiota with higher microbial diversity compared to healthy subjects, indicating that there could be more harmful bacteria or opportunistic pathogens in the oral cavity of patients with RA”. (Chen, B et al. Scientific Reports 2018; 8: 17126). It’s quite the same as the original article.
The reviewer could not understand “dysbiosis observed in RA and OA” derived from their old ages or the inflammatory disease condition even in the revised manuscript.
Author Response
Dear Reviewer,
despite not much data in the literature able to give final answers to your questions, we have tried to reply and add in the text your requests. You may find in the attached manuscript all the changes (highlithed in yellow).
Best regards.
Reply to the Reviewer number 2
The manuscript was improved. However, the authors could not properly respond to the reviewer’s comments. Moreover, the authors copied some of the phrases of original article.
Answer: Authors have tried to follow and address all the concerns raised by the Reviewer in the first revision. The suggestions and changes required have provided a huge improvement of the manuscript, and we would like to thank the Reviewer for this support. Despite the manuscript has been rewritten and changed in those parts requested by the Reviewer, and an additional table has been inserted as well as a further English revision by US researchers (dr Goswani and Parvizi) made, we are very sad to understand that the Reviewer is not fully satisfied. As the Reviewer knows, the field is not so full of references, and many discovers are still debated and to be furtherly clarified. This review aimed indeed not only to represent just an overview on what has been published on microbiota and arthritis, but also, and especially, to highlight controversies and future directions. Any effort has been made to underline pros and cons as well as doubts raising this field. So, we think any comment by the Reviewer is welcome, but we ask to the Reviewer to consider also the limitations and the biases which grips this field so far.
In line 110-112, authors copied the phrases of the original article “RA and OA had an oral microbiota with higher microbial diversity compared to healthy subjects, indicating that there could be more harmful bacteria or opportunistic pathogens in the oral cavity of patients with RA”. (Chen, B et al. Scientific Reports 2018; 8: 17126). It’s quite the same as the original article.
Answer: As the Reviewer can understand, there are few articles about Oral Microbiota and Artrhitis. The most important is probably that mentioned by the Reviewer and authored by Chen and others. Thus, it could be quite obvious that some sentences may be similar or having the same meaning. However, the phrase has been now rewritten (highlighted in yellow).
The reviewer could not understand “dysbiosis observed in RA and OA” derived from their old ages or the inflammatory disease condition even in the revised manuscript.
Answer: Some studies, especially in RA, are age-matched. So, we have tried to specified in the manuscript the role of microbiota independently by the age. Again, not many studies are present in literature, and conclusion may be indeed speculative and not fully conclusive.